# Effect of Nanosecond Pulsed Currents on Directions of Cell Elongation and Migration through Time-Lapse Analysis

**DOI:** 10.3390/ijms24043826

**Published:** 2023-02-14

**Authors:** Hayato Tada, Satoshi Uehara, Chia-Hsing Chang, Ken-ichi Yano, Takehiko Sato

**Affiliations:** 1Institute of Fluid Science, Tohoku University, Sendai 980-8577, Japan; 2Graduation School of Engineering, Tohoku University, Sendai 980-8577, Japan; 3Institute of Industrial Nanomaterials, Kumamoto University, Kumamoto 860-0862, Japan

**Keywords:** cell morphology, cell behavior, electric current, electric field, migration velocity, plasma medicine, discharge-simulated, chemical species free, agarose bridge, microchannel

## Abstract

It is generally known that cells elongate perpendicularly to an electric field and move in the direction of the field when an electric field is applied. We have shown that irradiation of plasma-simulated nanosecond pulsed currents elongates cells, but the direction of cell elongation and migration has not been elucidated. In this study, a new time-lapse observation device that can apply nanosecond pulsed currents to cells was constructed, and software to analyze cell migration was created to develop a device that can sequentially observe cell behavior. The results showed nanosecond pulsed currents elongate cells but do not affect the direction of elongation and migration. It was also found the behavior of cells changes depending on the conditions of the current application.

## 1. Introduction

In recent years, cold atmospheric pressure plasma (CAP) has been focused to apply in various fields, such as hydrogen conversion, waste treatment, surface modification, and medical application [1]. Plasma medicine is one of the applications of CAP and has a high potential for treatments, such as argon plasma coagulation, wound healing [2], angiogenesis [3], bacteria sterilization [4,5], and cancer inactivation [6] to name a few. CAP supplies a variety of physical and chemical stimuli, including UV rays, electric fields (EF), electric charges, excited atoms or molecules, chemically reactive species, and so on. Reactive oxygen and nitrogen species (RONS) in liquid media, which are generated by exposure to CAP, are currently regarded to be one of the key factors to inactivate cancer cells [7,8,9], and CAP at appropriate intensity can kill the cancer cell by inducing apoptosis, programmed cell death. This apoptosis killing by CAP is thought to decrease the burden on patients and receives considerable attention as the next cancer therapy. For CAP cancer therapy, direct treatment, such as a plasma jet [2], and indirect treatment, such as plasma-activated media (PAM) or plasma-activated ringer’s lactate solution (PAL), is developed [7,10]. In the case of direct treatment, it can supply both short-life RONS and long-life RONS [11]; however it also applies the electric factors of CAP, such as an EF or electric charges (current). The previous studies focused on the effects of RONS on cancer cells and the mechanism of apoptosis. The synergistic cytotoxic effect of nano-pulsed electric currents and RONS in CAP was also reported [12]. Leaving from the view of CAP, electrotaxis [13], electrochemotherapy [14], irreversible electroporation [15], or nanosecond electric field (nsPEF) electroporation [16,17,18,19] were reported as an effect of electric stimuli on cancer cells. Considering the CAP by high voltage-pulses, pulse duration should be 10 ns–1 μs to avoid the arc discharge transition so nsPEF electroporation partly shares the electrical condition. However, the strength of the electrical field seems much higher compared with CAP-producing conditions.

Our previous research focused on the plasma-simulated electrical stimuli and indicated the nanosecond pulsed current (nsPC) stimuli have a long-time scale effect on cellular responses, specifically, the human sarcoma cell line, HT-1080, changes its morphology to an elongated state without viability change. It was also indicated that actin fiber reorientation to one direction or stress fiber formation was induced at the same time [20]. An additional study clears the mechanism that Rac/Cdc42 protein signaling were activated by the nsPC, which triggers the intracellular ROS increase and continuous Ca^2+^ releasing or larger ATP production. Consequently, the cell changes its morphology to the elongating shape, generally called a mesenchymal state [21]. This phenomenon is unique as compared with previous findings on the effect of electrical stimuli on cancer cells. To achieve further clarification of the mechanism of this morphological transition, a time-lapse observation method is needed. This is because the time-lapse method can analyze or trace the individual cell morphological change and migration characteristics. Especially, the migration characteristics change is closely related to the cancer invasion or metastasis; thus, tracing the migration is necessary from the aspect of actual cancer treatment.

From these backgrounds, this study aims to provide a novel insight into the electrical aspect of CAP using the time-lapse method and customized algorithm analysis. For the first step, the new time-lapse experiment system with electrical stimuli was developed using narrow agarose bridges, a microchannel, and a stage-top incubator. This experiment system could isolate the electrical stimuli from chemical species made by electrolysis around electrodes, such as RONS. It also enables us to observe successive morphological change on HT-1080 with the application of nsPC. To analyze the migration characteristics with the mesenchymal transition for a long time, the novel automatic migration analysis method, which combines the cell nucleus live imaging and a customized algorithm was also developed. This automatic migration analysis ensures high accuracy compared with human-tracking results and dramatically made cell tracking easier. Finally, we applied this new analysis algorithm to an image sequence taken by a new time-lapse observation system and analyzed the migration of characteristics with the mesenchymal transition by nsPC.

## 2. Results and Discussion

### 2.1. nsPC Stimulation Change HT-1080 Morphology with Stress Fiber Formation

In this section, all experiments were held with the nsPC-800 condition. Firstly, we tried to observe the nsPC effect on human sarcoma cells using the time-lapse method. The cells began to change their morphologies 10 to 11 h later from nsPC application. Figure 1 shows the cell images of the control and nsPC in 0, 12, and 24 h. Compared with the control condition, the cells were elongated and showed the protrusion formation indicated by yellow arrows. The number of elongated or protrusion-formed cells was increased with the duration of exposure to the nsPC. The cell viability at 24 h was confirmed using Hoechst33324 and propidium iodide and calculated to be 99.0 ± 0.4% in the control group and 99.2 ± 0.7% in the nsPC group. This result is different from the previous study on the effect of weak nano-second pulsed electric field (nsPEF), which showed a field strength of 500 V/cm and had the potential to selectively kill the cancer cells [22,23]. This comparison suggests the distinctive effect of nsPC.

Immunofluorescent was performed to show the α-Tubulin and actin filament alternation in the nsPC group. As shown in Figure 2a, there are no differences about α-Tubulin (green) between the control group and the nsPC group. However, using the higher resolution, the number of cells with actin filament reorientation and stress fiber formation appeared to be increased with the application of nsPC, which is shown in Figure 2b. In addition, several cells seemed to exhibit a high-actin concentration on the edge of the cell membrane at 24 h instead of showing the actin fiber reorientation or stress fiber formation. To verify our observations on the alterations in fiber formation, quantitative analysis of individual fibers with/without nsPC is essential. In our future investigation, we will explore the possible relationship among the extent of fiber formation, the directions of individual fibers, and the direction of electric current.

Previous research demonstrated sarcoma-type cancer cell lines, such as HT-1080, exhibit several morphological modes, including ameboid state, mesenchymal state, and epithelial state [24,25]. In this study, we observed cell elongation and protrusion, both which started at 10–11 h after nsPC stimulation and became evident at 12 and 24 h, as shown in Figure 1. These morphological changes are very similar to the characteristics of the mesenchymal state, which is in good agreement with our previous research [21]. This morphological transition seems to be related to the epithelial–mesenchymal transition (EMT) and ameboid–mesenchymal transition (AMT), the reversible processes to gain the appropriate characteristics for cancer migration or metastasis [26]. Previous research also reported that the Rho sub-family, one of the Ras super-family, is related to the EMT or AMT [27,28] and our previous study actually showed Rac/Cdc42 activation is upstream of this morphological transition [21]. This Rho sub-family activation also has the potential on changing the cell migration characteristics; therefore, migration analysis on this phenomenon is definitely needed.

### 2.2. Accuracy of PTV Method Compared with Manual Method

Because cell morphological transition with nsPC took a long time to appear, we needed the automatic cell migration analysis method to deal with a large number of cells for a long time. In this study, we apply the PTV method, one of the fluid motion analysis methods, on the cell by combining the nucleus staining and time-lapse observation with an appropriate interval time and customized algorithm. To ensure the high accuracy of the PTV method, we compare it with the calculated results of the manual method and the PTV method in the same control sample. Two methods of analysis were performed every 15 min for all cells in the observed area, and Figure 3 shows the summarized results. Here, the sample number of “n” denotes the total cell count at the defined 8 h. Figure 3a shows the result of the cell migration direction analysis. The migration ratio of cells in each direction was obtained by (cell count in one direction)/(total cell count in all directions). It is obvious that cells were randomly migrated, and compared with these two methods, the PTV method could catch the detailed ratio difference as similar to the manual method. Figure 3b exhibits the results of migration velocity in each direction and the cell migration was almost uniform in each direction and each time. The results of velocity by PTV method are also in good agreement with the manual result.

To quantify the accuracy of the PTV method, we obtained the capture rate defined by the ratio of the total cell count by PTV method at the defined 8 h and the total cell count by manual method at the defined 8 h. The capture rate was calculated in three control samples, and the results are shown in Table 1. This table showed over 90% of cells in each image were recognized by this algorithm, and this number was enough to analyze the cell migration characteristics. The correlation coefficient between the results of the PTV method and the manual method was also obtained to quantify the accuracy. The correlation coefficient on the migration ratio of cells in each direction at every 8 h was 0.89 ± 0.04 and that on average, the velocity was 0.87 ± 0.06. Hence, we evaluated that the PTV method developed in this study has enough ability to analyze the migration characteristic automatically with high accuracy.

### 2.3. Effect of nsPC on Cell Migration

The alternation of cell migration characteristics with the morphological transition was analyzed using the PTV method. In condition nsPC-800, the number of elongated cells is still small for time-lapse observation. For the higher possibility to observe the cell elongation phenomena, we elevated the maximum current to 1000 mA, and this condition was called nsPC-1000. Figure 4 shows the images taken after 24 h in the conditions of control and nsPC-1000. Compared with the results of the nsPC-800 in Figure 1, the number of elongated cells was increased in nsPC-1000, as we hypothesized.

The calculation results of migration characteristics in the control and the nsPC-1000 are shown in Figure 5. Compared with the migration ratio of the control and the nsPC-1000 shown in Figure 5a,c, there are no significant changes in both conditions and showed random migration in each period. The calculation results of average velocity are shown in Figure 5b,d, and there were also no significant differences. Many previous studies on DC electrical stimuli have clarified the impact on the cell, such as galvanotaxis [13,29,30]; on the other hand, we demonstrated that galvanotaxis or migration acceleration was not introduced in the case of ultra-short electrical stimuli application.

## 3. Materials and Methods

### 3.1. Cell Culture

The human sarcoma cell line, HT-1080, was obtained from the Japanese Collection of Research Bioresources Cell Bank (Osaka, Japan). Cells were grown in minimum essential medium eagle (Sigma-Aldrich, St. Louis, MO, USA) supplemented with 10% fetal bovine serum (Thermo Fisher Scientific, Waltham, MA, USA) and 1% penicillin/streptomycin (Nacalai Tesque, Kyoto, Japan). Cells were cultured under humidified conditions with 5% CO_2_ at 37 °C. For passage, 0.25 *w*/*v*% trypsin/1 mM EDTA solution (Fujifilm Wako Pure Chemicals, Osaka, Japan) was used. Cells with passage numbers 3–7 were used in the described experiments.

For the experiment, 200 µL of cell suspension were seeded in µ-Slide I (ibidi, Gräfelfing, Germany, size of the flow channel is 5 mm × 50 mm × 0.4 mm) at a concentration of 6.0–7.0 × 10^5^ cells/mL and were allowed to attach to the bottom of the slide for 6 h incubation. After cell attachment, 0.9 mL of fresh medium was added to fill each reservoir, and a total of 2.0 mL media was loaded in each slide.

### 3.2. Experimental Setup with nsPC

We used the same concept of the experimental setup as the previous work for the application of nsPC [20,21]. In this study, we developed a new device that was able to be installed on the stage of the fluorescence microscope. This new device employed a microchannel to suppress the evaporation of the culture medium, which also significantly reduced the space requirement in our experimental setup. In addition, this new device successfully achieved sufficient electrical insulation when nsPC was applied in the incubator. The schematic illustration is shown in Figure 6. A cultivated slide was set in a stage-top incubator and kept the condition of humidified atmosphere with 5% CO_2_ at 37 °C. The outside and inside of the incubator were connected by the agarose bridges. For agarose bridge construction, a 6 mm diameter glass tube and a 6 mm diameter silicon tube were combined and used as a container of agarose. Then, 2 wt% of agarose (Sigma-Aldrich, No. A7002) was solved in PBS (Gibco^TM^, Billings, MT, USA, pH 7.2) with heating and poured into the container. After cooling, one end of the agarose bridge was connected to a phosphate-buffered saline (PBS) solution tank into which an Ag/AgCl (Nilaco Corp., Tokyo, Japan, AG-401325) electrode was inserted, and the other end was connected to a μ-Slide filled with culture medium to avoid the influence of chemicals generated by the electrolysis around the electrode [29].

Nanosecond pulsed current (nsPC) was generated by power sources designed and manufactured in our laboratory. The electrical circuit of power sources is shown in Figure 1. These power sources are controlled by a function generator (WF1974, NF Corporation, Yokohama, Japan).

Figure 7 shows the waveforms of applied voltage (solid line) and discharge currents (dotted line) used in this study. Briefly, we distinguish the condition by maximum current, around 800 mA (named nsPC-800) or around 1000 mA (named nsPC-1000). In both conditions, we set 100 Hz for frequency.

During the experiment, maintaining a stable temperature and pH of culture media is essential; thus, these parameters were monitored. The results of monitoring are shown in Table 2 and suggest the joule heating effect and pH changing by the application of nsPC is negligible. The condition of nsPC-800 was used for viability check and actin filament reorganization analysis; otherwise, nsPC-1000 was used for migration analysis.

### 3.3. Cell Staining

Cells were stained for three purposes, viability check, α-tubulin and stress fiber imaging, and migration analysis.

For the viability checking, 1 µg/mL Hoechst 33342 (Fujifilm Wako Pure Chemicals) and 1 µg/mL propidium iodide (Thermo Fisher Scientific) were used after the experiment.

To visualize α-tubulin, cells were stained by the immunofluorescence method [21]. At first, cells were fixed in 4% paraformaldehyde dissolved in PBS and then, permeabilized in 0.2% Trion X-100 (Sigma-Aldrich, T8787). Next, cells were blocked with 2% bovine serum albumin and subsequently treated with the antibody against α-Tubulin (Sigma-Aldrich). After washing with PBS, cells were incubated with goat anti-mouse IgG-Alexa Flour 488 (Thermo Fisher Scientific) and mounted in a Vectashield mounting medium (Vecter Laboratories, Newark, CA, USA). For visualizing the stress fiber formation, rhodamine-labeled phalloidin (Cosmo Bio, Tokyo, Japan) was used.

To analyze the effects of nsPC on migration, cells were treated before the experiment. After 6 h of incubation for cell attachment, the culture media was changed to the 200 µL of fresh media with 0.2 μL of NucSpot^TM^ Live 488 (Biotium, Fremont, CA, USA) Nuclear Stain and 15 µM/mL of Verapamil (Biotium). After 10 min incubation, 0.9 mL of fresh media with 15 μM/mL of Verapamil was added to each reservoir and incubated an additional 50 min. Hence, in the migration analysis, the total incubation time before the experiment is 7 h.

### 3.4. Microscopy and Time-Lapse Analysis

The Axio Observer D1 (Carl Zeiss, Oberkochen, Germany) and the software ZEN2.5 (blue edition) (Carl Zeiss Microscopy GmbH, Germany) was used for fluorescence and differential interference contrast (DIC) microscopy in a single image and time-lapse analysis. The cells were imaged every 3 min for only a DIC time-lapse. In the case of time-lapse by both fluorescent and DIC, the interval time was set to 15 min for avoiding phototoxicity. The time-lapse observation was continued for 24 h with 20× magnification.

### 3.5. Automatic Cell Migration Analysis by Algorithm

Image sequences taken by the time-lapse method are used for migration analysis. To determine the velocity and direction of migration in each cell, we need to determine the center coordinate of the cell. The way to determine the center coordinate in the previous study is to calculate the best-fit ellipsis or center of gravity from the cell contour [30,31,32]. However, there are two difficulties in yielding the cell contour automatically using a DIC image. One is setting the contour in the unclear part of the cell, and the other is setting the border of the connected cell. To overcome these obstacles, the gravity center of the nucleus fluorescent area was used as the center coordinate of the cell in this study.

The custom algorithm was developed by written python for cell tracing. First, the fluorescence image of the nucleus was binarized, and the noise was removed by filtering the area of fluorescence. To link the cell in each image, a particle tracking velocimetry (PTV) method, one of the velocity vector analysis methods, was applied. In the PTV method, when the flame rate of the camera is high enough compared with a particle velocity in the fluid, a velocity vector of a luminescent particle can be obtained by only linking the closest luminescent particle. In this study, the position of the center of gravity of the fluorescent region of the nucleus was treated in the same way as the luminescent particles in the PTV method, and the vector data of the cells were analyzed and used to explain cell migration (Figure 8a). The flame rate of the cell migration in this study was four images/hour, which was high enough compared with the scale of cell migration.

The vector data of cells were sorted in four directions as shown in Figure 8b, and then, the numbers of cells in each direction and the total number of cells were counted. The migration ratio of cells was obtained by dividing the cell count in one direction by the total cell count in all directions. The average velocity in each direction was also calculated from vector data of cells. Migration direction and the average velocity of cells were regarded to be the characteristics of cell migration.

To ensure the accuracy of the PTV method, vector data of cells were also obtained by manually tracing the coordinate of the nucleus gravity center and were used as a comparison target. This manual analysis method is called a manual method in this study.

### 3.6. Statistical Analysis

All experiments were repeated at least three times. The data were expressed as mean ± SD.

## 4. Conclusions

This paper aims to provide a novel insight into the electrical aspect of CAP, combining the time-lapse method, fluorescence imaging, and customized algorithm analysis. We succeeded in developing a new time-lapse experiment system with electrical stimuli, which enables us to isolate the electrical stimuli from chemical species made by electrolysis around electrodes. Additionally, we also succeeded in developing the automatic migration analysis method, which combines the cell nucleus live-imaging and a customized algorithm. This automatic migration analysis ensures high accuracy compared with human-tracking results and dramatically made cell tracking easier. Using these new systems, we investigate the effect of the nsPC on cell morphology or migration and received a remarkable suggestion that the nsPC has potential for morphological transition; however, it does not affect the migration characteristics. These results are different from the effect of DC or nsPEF and suggest the nsPC’s effect on the cell.

## Figures and Tables

**Figure 1 ijms-24-03826-f001:**
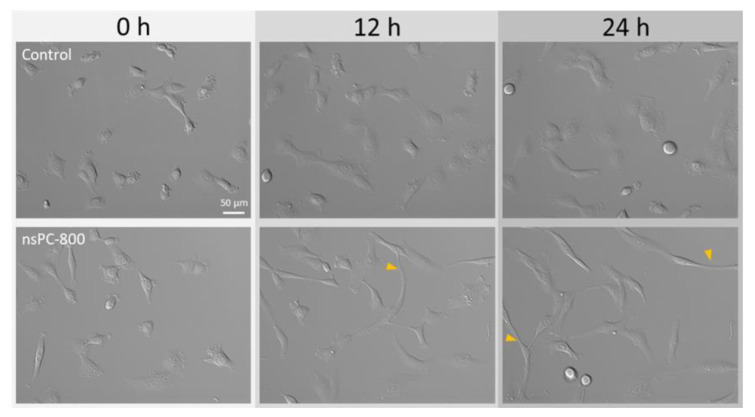
Representative time sequence of DIC images in time-lapse observation at ×20 magnification.

**Figure 2 ijms-24-03826-f002:**
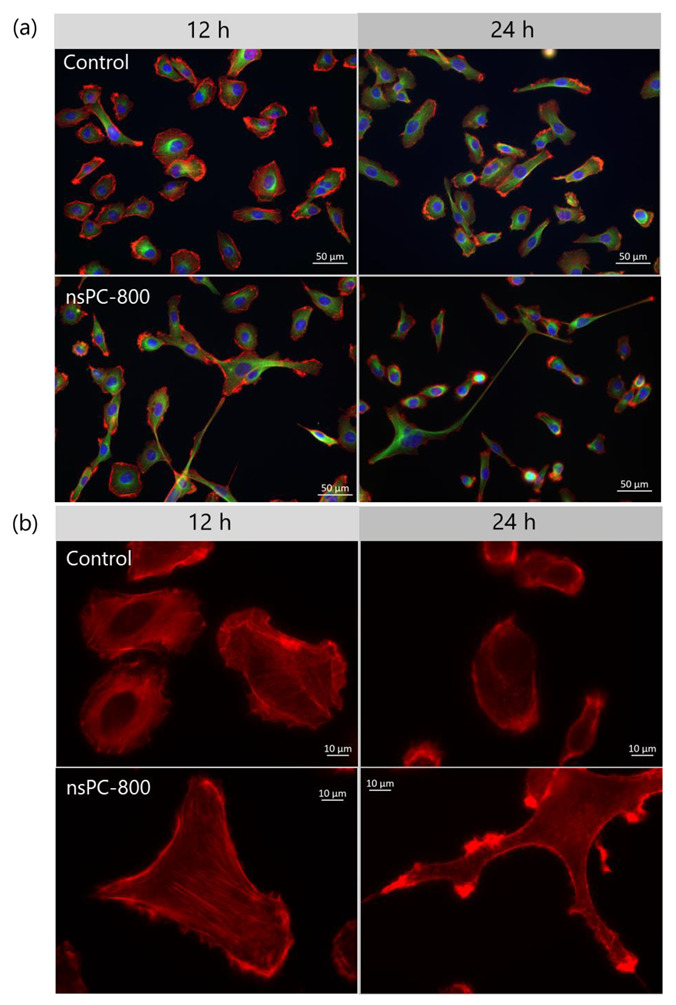
Comparison of cell morphology in immunofluorescence. (**a**) Cell images in control (upper) and nsPC-800 (lower) are taken at 12 h and 24 h. Immunofluorescence images of Hoechest 33342 (blue, nucleus), anti-α-tubulin antibody (green, α-tubulin), and rhodamine-labeled phalloidin (red, actin) were merged at ×20 magnification. (**b**) Actin fiber reorientation and stress fiber formation were observed at ×63 magnification in control (upper) and nsPC-800 (lower).

**Figure 3 ijms-24-03826-f003:**
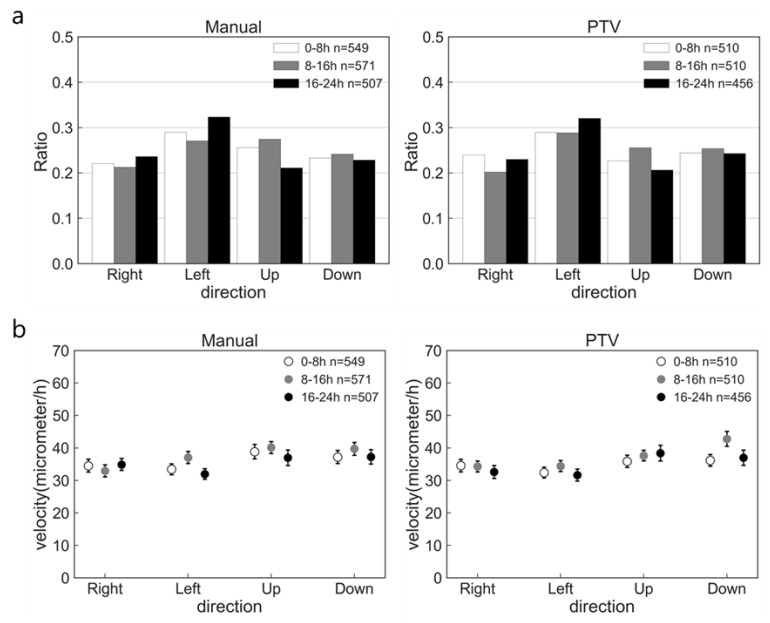
Comparison of migration analysis results by manual and PTV methods in one control sample. (**a**) Ratio of cell number in each direction. (**b**) Average velocity in each direction and each time. The error bar shows the standard error. The “n” denotes the total cell count at the defined 8 h.

**Figure 4 ijms-24-03826-f004:**
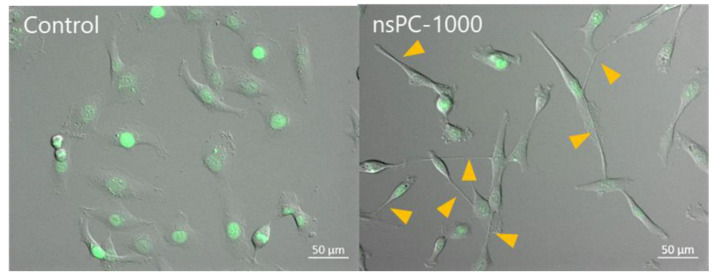
Images were taken after the 24-h experiment with control and nsPC-1000. Cells were treated with NucSpot^TM^ Live 488 and the nucleus shows green fluorescence. Yellow arrows show the elongation part of cells in the nsPC-1000.

**Figure 5 ijms-24-03826-f005:**
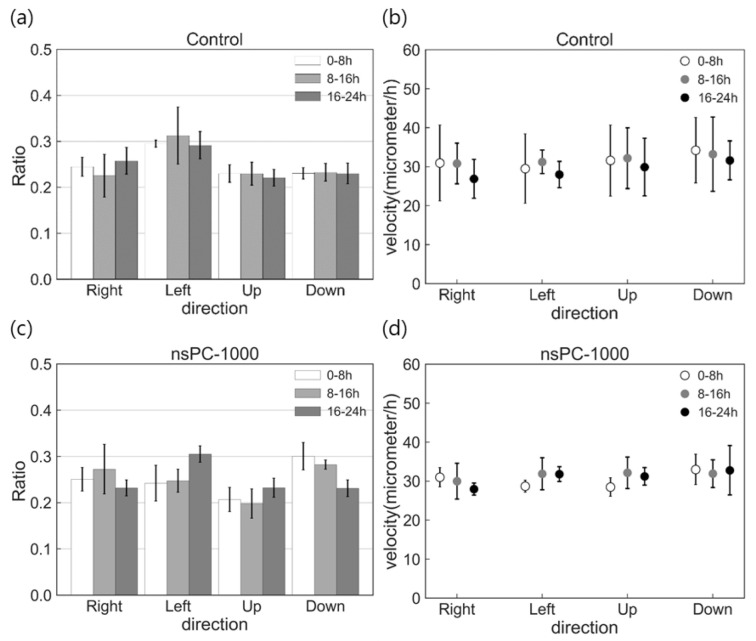
Cell migration characteristics were investigated by calculation of migration ratio and average velocity. (**a**,**b**) show the results of direction ratio and average velocity in the control condition and (**c**,**d**) in the nsPC-1000.

**Figure 6 ijms-24-03826-f006:**
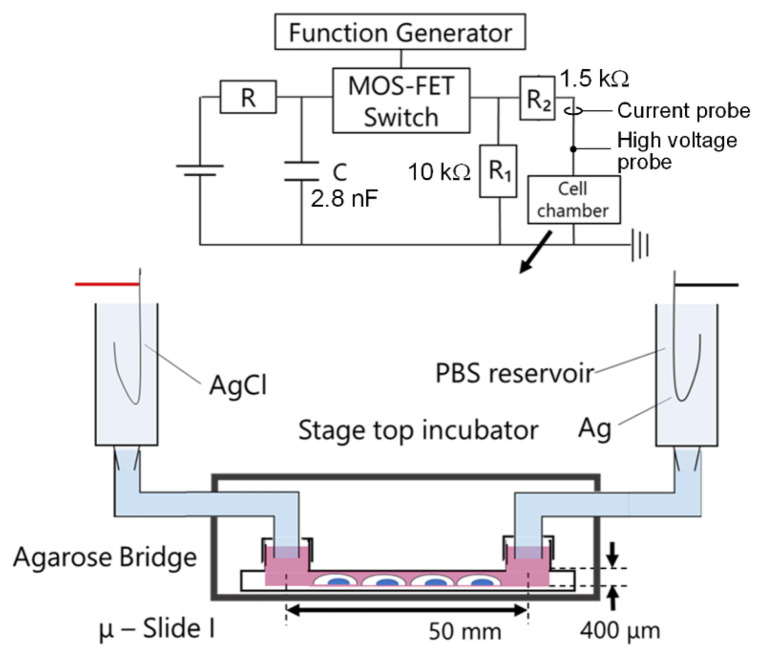
The schematic illustration of the stimulation system and power supply circuit.

**Figure 7 ijms-24-03826-f007:**
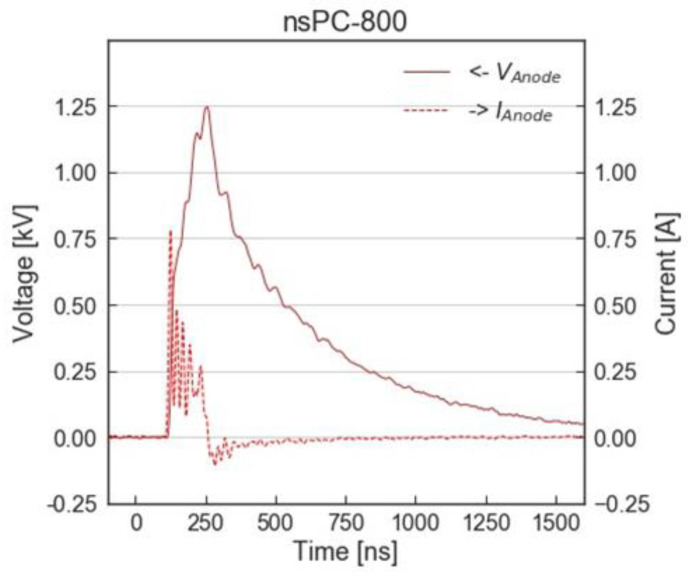
The waveforms of applied voltage and pulsed current in nsPC-800.

**Figure 8 ijms-24-03826-f008:**
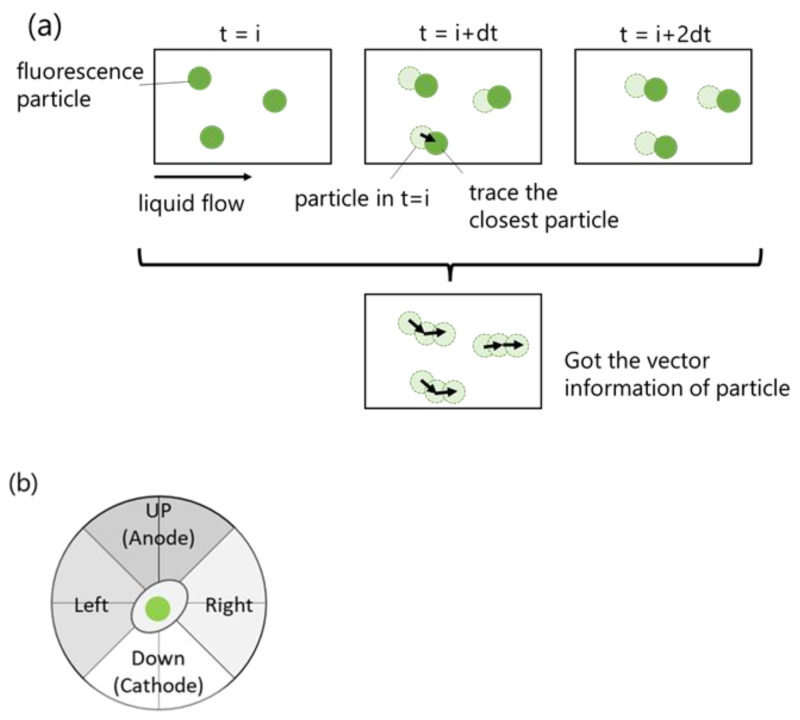
(**a**) Simple image of PTV method. (**b**) Cells were sorted into four directions.

**Table 1 ijms-24-03826-t001:** Capture rate in PTV.

	0–8 h	8–16 h	16–24 h
**Capture ratio in PTV (vs. manual) [%]**	93.0 ± 0.7	90.1 ± 5.0	96.2 ± 5.2

**Table 2 ijms-24-03826-t002:** Temperature and pH were monitored in each condition. The effect of joule heating and pH are negligible in every condition.

Temperature (°C)	Start	24 h Later
**Control**	35.0 ± 0.9	34.2 ± 0.5
**nsPC-800**		34.4 ± 0.7
**nsPC-1000**		34.8 ± 0.3
**pH**	**Start**	**24 h Later**
**Control**	7.7 ± 0.1	7.3 ± 0.1
**nsPC-800**		7.4 ± 0.3
**nsPC-1000**		7.6 ± 0.1

## Data Availability

The data that supports the findings of this study are available in the master thesis of Hayato Tada, the graduate school of engineering, Tohoku University in 2021.

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
