# Peer review of "Effect of Nanosecond Pulsed Currents on Directions of Cell Elongation and Migration through Time-Lapse Analysis"

_ijms, 2023, doi:10.3390/ijms24043826_

Round 1
Reviewer 1 Report
The study provided a novel aspect of cold atmospheric pressure plasma (CAP) usage for investigating the effect of nanosecond pulsed current (nsPC) stimuli on cell elongation and migration. By using time-lapse method and customed algorithm analysis, the authors can obtain the images of the human sarcoma cell line HT-1080 with its morphology change and migration status. Compared with the previous researches, the authors in this study implemented new algorithm, which significantly make the cell tracking easier with relatively high accuracy compared to human-tracking method. Also, by comparing the different level of nsPC, the authors suggested that the nsPC potentially influenced the morphological transitions but may not alter the migration characteristics, which helps future investigation on the study in the field.
With that being said, this paper provided good practice guidance for investigating the nsPC effect on cell morphology and migration with their automatic algorithms. However, there are two major issue and several minor issues that need address or improve:
1. The description of the custom algorithm is too simple to help reader understanding the novelty for the method. For example, the program can be visualized for presentation considering the confidential information cannot be presented. Also, the verification of the method is hard to track due to the same reason.
2. The devices and the experiment design are very similar to the previous literatures such as 20, 21 from the same authors, which lowers the novelty of the paper. The author should think about highlighted the improvement parts more.
Minor:
1. Line 28-29, the sentence is not clear.
2. Line 55-56, it is better to say “This phenomenon is unique compared with previous finding on the effect of electrical stimuli on cancer cell”.
3. Line 96, “put” can be described more in more details for preventing the side effect.
4. Line 215, Section title should be “Accuracy of PTV method compared with manual method”.
5. Line 217” larger” should be “large”.
Author Response
Dear Reviewer 1,
Thank you for waiting for our response. We have attached our response to the reviewers' comments and the revised manuscript for #IJMS-2167465 for your review.
Sincerely yours,
Takehiko Sato

Reviewer 2 Report
The manuscript entitled “Effect of nanosecond pulsed currents on directions of cell elongation and migration through time-lapse analysis” submitted to IJMS in view of publication (corresponding author T. Sato) addresses the effects of nanosecond pulsed currents on cell behaviour. Cold atmospheric pressure plasma is applied to cells and time-lapse observation is performed to investigate cell migration and cell elongation using image recognition algorithms. Nanosecond electrical pulses applied to sarcoma cells have potential to lead to a morphological transition but they have no effect on cell migration. The manuscript brings an interesting contribution to cell morphology control and I suggest publication upon taking into account some minor corrections that are indicated below.
Line 42 & lines 54-57 : It is interesting to indicate if the cell behaviour upon application of nanosecond electric pulses (or other electric stimuli) is different for normal cells and for cancer cells.
Line 66 : Indicate which are the electrolysis compounds, which are their effects on the cells, and which is the interest to isolate them.
Lines 94-100 : Give all physical and chemical composition parameters that describe the agarose bridge composition. Demonstrate clearly that this method is efficient to separate the electrolysis compounds.
Figure 1 : Indicate all values of the physical parameters in the electrical circuit (time dependences of the voltages of the sources, the values of the resistors and of the capacitor). Indicate the width of the 50 mm-long cell slide.
Figure 2 : Indicate at which points of the electrical circuit (Fig. 1) were performed the measurements of the time dependences of the current and of the voltage. Relate the time constant of the discharge voltage with the values of the electrical components RC and indicate if cell slide contribute to that. Why the current and the voltage dependences are not similar?
Line 127 : Add bibliographic reference describing the method of immunofluorescence.
Line 143 : Add bibliographic reference for the DIC microscopy.
Line 154 : Explain clearly what is the algorithm that you developed in order to perform cell image recognition. Indicate which are the uncertainties in cell position measurement by microscopy and from image recognition algorithm. Fix some redundant typesetting errors regarding the algorithm: costumed & customed.
Line 185 : Conclude clearly if ns electrical pulses may induce morphology changes selectively on cancer cells or no.
Line 194-195 : Indicate quantitatively the effects of actin fiber reorientation and stress fiber formation.
Line 202-213 : Indicate the timescales for morphological transitions for cancer cells.
Line 221 : Define clearly the ratio of cell numbers.
Figure 6 : Define in the text of the manuscript the quantities indicated on the legends of the plots “n=…”.
Line 245 : Define the quantity capture rate.
Line 250 : Define the quantity correlation coefficient between the values of the measurements.
Lines 279-283 : Indicate relevant bibliographic references for previous researches on electrical stimuli.
Author Response
Dear Reviewer 2,
Thank you for waiting for our response. We have attached our response to the reviewers' comments and the revised manuscript for #IJMS-2167465 for your review.
Sincerely yours,
Takehiko Sato
